# Administration of an Acidic Sphingomyelinase (ASMase) Inhibitor, Imipramine, Reduces Hypoglycemia-Induced Hippocampal Neuronal Death

**DOI:** 10.3390/cells11040667

**Published:** 2022-02-14

**Authors:** A Ra Kho, Bo Young Choi, Song Hee Lee, Dae Ki Hong, Beom Seok Kang, Si Hyun Lee, Sang Won Suh

**Affiliations:** Department of Physiology, College of Medicine, Hallym University, Chuncheon 24252, Kangwon-do, Korea; arakho136@naver.com (A.R.K.); bychoi@hallym.ac.kr (B.Y.C.); sshlee@hallym.ac.kr (S.H.L.); zxnm01220@gmail.com (D.K.H.); ttiger1993@gmail.com (B.S.K.); irish8281@naver.com (S.H.L.)

**Keywords:** hypoglycemia, sphingolipid, acidic sphingomyelinase, ceramide, imipramine, neuron death, cognitive dysfunction

## Abstract

Severe hypoglycemia (below 35 mg/dL) appears most often in diabetes patients who continuously inject insulin. To rapidly cease the hypoglycemic state in this study, glucose reperfusion was conducted, which can induce a secondary neuronal death cascade following hypoglycemia. Acid sphingomyelinase (ASMase) hydrolyzes sphingomyelin into ceramide and phosphorylcholine. ASMase activity can be influenced by cations, pH, redox, lipids, and other proteins in the cells, and there are many changes in these factors in hypoglycemia. Thus, we expect that ASMase is activated excessively after hypoglycemia. Ceramide is known to cause free radical production, excessive inflammation, calcium dysregulation, and lysosomal injury, resulting in apoptosis and the necrosis of neurons. Imipramine is mainly used in the treatment of depression and certain anxiety disorders, and it is particularly known as an ASMase inhibitor. We hypothesized that imipramine could decrease hippocampal neuronal death by reducing ceramide via the inhibition of ASMase after hypoglycemia. In the present study, we confirmed that the administration of imipramine significantly reduced hypoglycemia-induced neuronal death and improved cognitive function. Therefore, we suggest that imipramine may be a promising therapeutic tool for preventing hypoglycemia-induced neuronal death.

## 1. Introduction

Diabetes is one of the most common metabolic diseases currently affecting public health, with growing concern about the increase in diabetes patients globally. Pouya Saeedi et al. noted that the global diabetes prevalence in 2019 was assumed to be 9.3% (463 million people), and it is expected to increase to 10.2% (578 million) by 2030 [1]. Type 1 and type 2 diabetes patients are routinely treated with insulin or insulin-releasing drugs in order to reduce their blood glucose levels. In recent decades, many diabetes patients have experienced hypoglycemia while attempting to tightly control their blood glucose to prevent diabetic complications [2]. According to the Centers for Disease Control and Prevention (CDC), diabetic patients may experience low blood sugar as often as once or twice a week in the case of severe hypoglycemia (below 54 mg/dL). Additionally, according to Alexandria Ratzki Leewing et al., as published in the 2018 BMJ journal, severe hypoglycemia was reported by 41.8% of all respondents, at an average rate of 2.5 events per person/year [3]. Devices that continuously monitor blood glucose levels and optimize the delivery of insulin or insulin-like drugs have been successful in treating diabetes. However, these approaches are not always successful in the complete prevention of hypoglycemia.

Severe hypoglycemia occurs when blood glucose levels decrease below 35 mg/dL. When blood glucose falls further below 20 mg/dL, irreversible brain damage can take place in vulnerable brain regions. According to previously published studies, repeated and severe hypoglycemia in humans induces neuronal damage in the cortex, particularly the insular cortex, hippocampus, caudate, and putamen [4]. This neuronal damage impairs cognitive function in diabetic subjects [5,6,7], and severe hypoglycemia can lead to coma, seizure, cognitive impairment, and even death. Hypoglycemia itself can cause neuronal damage, but our lab has demonstrated that glucose reinstatement can cause an increase in neuronal death after hypoglycemia. Thus, we called it “hypoglycemia/glucose reperfusion injury” after hypoglycemia [8]. We have demonstrated that abrupt changes in blood glucose concentrations can cause the disturbance of several cellular programs that promote neuronal death, lead to the destruction of the blood–brain barrier (BBB), enhance the production of reactive oxygen species (ROS), and cause excessive zinc release and inflammation known to be mediated by the overactivation of microglia [9,10,11,12].

Sphingomyelin (SM) is a type of sphingolipid found in animal cell membranes, and it plays an important structural and functional role in neurons and is associated with many signaling pathways. Sphingomyelin is hydrolyzed to ceramide and phosphorylcholine. The hydrolysis of sphingomyelin present in the plasma membrane is catalyzed by the neutral sphingomyelinase (NSMase) and sphingomyelin present in the membranes of endosomes and lysosome is hydrolyzed by the enzyme acid sphingomyelinase (ASMase) [13].

The acid sphingomyelinase is a soluble lysosomal glycoprotein that acts in the breakdown of sphingomyelin to ceramide and phosphorylcholine. ASMase is ubiquitously expressed in almost every cell type, especially the endo-lysosomal compartment. However, under certain conditions, it can translocate to the plasma membrane, presumably at the outer leaflet [14]. ASMase is known to play a role in neural pathophysiology, as its activity has been linked to depression, BBB disruption, and ROS production, and it is activated by various stimuli such as lipopolysaccharides (LPS), oxidative stress, excessive cationic toxicity, lipid and tumor necrosis factor α (TNF-α), etc. [15,16,17].

Ceramide, a core constituent of all complex sphingolipids, plays an important role in numerous fundamental cellular processes, including the modulation of membrane permeability [18], cellular differentiation, growth, apoptosis [19], stress signaling [20], and inflammation [21]. As a neutral lipid, ceramide can be generated by ASMase and form ceramide-rich membrane platforms, which involves the induction of apoptosis and growth inhibition and other cellular responses [22].

Despite growing interest in ASMase/ceramide signaling in neurological diseases, no clear association has been established that would indicate a role of ASMase and ceramide-mediated neurodegenerative disease in pathological states [23]. Many researchers have already confirmed that the activation of ASMase and the increase in ceramide lead to ROS production, excessive inflammation, and severe neuronal death in various brain diseases such as Alzheimer’s disease, cerebral ischemia, multiple sclerosis and Parkinson’s disease [24,25,26,27].

Imipramine is a tricyclic antidepressant used to treat depressive mood disorders via inhibition of serotonin reuptake [28,29]. In addition to its antidepressant effects, imipramine has also shown a variety of effects on cell proliferation and the inhibition of apoptosis and inflammation, including an elevation of neurotrophic factors and exerting cyto- and neuroprotective effects [30,31,32]. Importantly, imipramine is also widely known as an ASMase inhibitor via an indirect functional mechanism. In general, ASMase is bound to the intracellular lysosomal membrane and protected from inactivation due to the fact of protein degradation. However, imipramine isolates ASMase from the inner lysosomal membrane, which can cause ASMase to be deactivated by protein degradation [33]. These various pharmacological characteristics of imipramine provide a rationale for this study to test the potential of these drugs to inhibit hypoglycemia-induced neuronal death.

The present study sought to determine the effect of ASMase and ceramide on hypoglycemia-induced neuronal death. We also demonstrated the neuroprotective effects of the ASMase inhibitor imipramine.

## 2. Materials and Methods

### 2.1. Ethics Statement

This study was conducted in accordance with the strict guidelines of the Institutional Animal Studies Care and Use Committee of the Hallym University in Chuncheon, Korea (Protocol no. Hallym-R1(2020-8)). Animal sacrifice was carried out using isoflurane anesthesia, and we attempted to cause minimal pain and distress.

### 2.2. Experimental Animals

Adult male Sprague–Dawley rats (8 weeks old, 250–350 g, DBL Co., Eumseong, Korea) were used. The animals were managed in a constant temperature- and humidity-controlled environment (22 ± 2 °C, 55 ± 5% humidity, and a 12-h light:12-h dark cycle) and provided feed (Purina, Gyeonggi, Korea) and water freely. This research was conducted in accordance with the Animal Research: Reporting in Vivo Experiments (ARRIVE) guidelines.

### 2.3. Animal Surgery and Severe Hypoglycemia Induction

The hypoglycemia model used in this study was a severe hypoglycemia model (HG). Animals were fasted for 15–16 h before the induction of hypoglycemia [12,34]. After overnight fasting, animals were injected with insulin into the abdominal cavity at a concentration of 10 U/kg. Hypoglycemia surgeries were conducted under a tightly controlled environment for breathing and anesthesia with a small rodent respirator (Harvard Apparatus, South Natick, MA, USA). First, the rats were anesthetized with 2–3% isoflurane in 70% NO_2_/balanced O_2_ via nose cone until an isoelectric period (iso-EEG) appeared. A catheter was inserted in the femoral vein for glucose reperfusion after hypoglycemia induction and inserted into the femoral artery for continuous blood pressure monitoring. When inducing hypoglycemia, observations of animal brain electroencephalogram (EEG) signals are important, and for EEG measurement, we inserted needles through two small holes into the skull bilaterally and placed a reference needle under the neck muscle. We continuously monitored blood pressure and EEG changes through the BIOPAC software system (BIOPAC System Inc., Santa Barbara, CA, USA). We kept the central temperature of the animals at 36.5–37.5 °C and measured their blood glucose level at 30-min intervals to confirm hypoglycemia. After insulin administration, the blood glucose level dropped, and after approximately 2~2.5 h, the glucose level dropped below 10mg/dL, inducing severe hypoglycemia in the animal model used in this study. When severe hypoglycemia is triggered, the EEG signal enters the iso-EEG, flattening the waves, and arterial blood pressure rises to between 130 and 200 mmHg during the isoelectric time. At that time, we lowered anesthesia to 0.5–0.75% isoflurane, changed the ratio of the oxygen and nitrogen to 7:3, and then maintained it for 30 min. Hypoglycemia-induced neuronal death is directly related to the duration of the iso-EEG period. After 30 min of this period, 25% glucose was injected into the femoral vein for 2 h (2.5 mL/h, i.v.) under 2% isoflurane to restore the rat from low blood glucose levels to normal levels [34,35].

### 2.4. Injection of Imipramine (IMP)

The rats were divided into 4 groups to determine the effects of imipramine on neuronal death after hypoglycemia: sham (vehicle, IMP (Sigma-Aldrich, Munich, Germany)) and hypoglycemia (vehicle, IMP). The sham group was injected with 0.9% saline, and after hypoglycemia induction, the animals were given an intraperitoneal injection of IMP (10 mg/kg, i.p.) once per day for 7 days.

### 2.5. Brain Sample Preparation

To confirm the neuroprotective effects of imipramine, animals were sacrificed 1 week after hypoglycemia using urethane (1.5 g/kg, i.p.) in order to anesthetize the animals sufficiently. The animals’ blood was removed by perfusion with 500–600 mL of 0.9% saline and then immediately perfused with 600 mL of 4% formaldehyde (FA) to fixation. After perfusion, we removed the brain and immersed it in the same formaldehyde (FA) for 1 h to perform post-fixation. After that, we immersed the brain in a 30% sucrose solution for 2–3 days for cryoprotection. When the brain is initially immersed in 30% sucrose, buoyancy causes the brain to float. Afterward, sucrose solution permeates into the brain tissue, and the brain sinks to the bottom 2~3 days later. After being fixed with formaldehyde, there was a pretreatment process of softening the brain tissue that prevents the tissue from breaking during cryosection. Following this, it was frozen with a freezing medium, placed on a cryostat, cut to a 30 µm thickness, and stored in a storage solution.

### 2.6. Enzyme Activity Assay for Sphingomyelinases (SMase)

The activities of acidic or neutral SMases were assessed, as described previously [36]. Briefly, brain samples suspended in appropriate SMase assay buffers (acidic SMase buffer: 250 mM sodium acetate, 0.2% Triton X-100, pH 4.5; neutral SMase buffer: 20 mM HEPES, 0.2% Triton X-100, pH 7.4) were incubated with 5 nmol of C12-sphingomyeline for 20 min at 37 °C. The reaction was stopped by the addition of CHCl_3_: CH_3_OH (2:1, *v*/*v*) and the organic phases were removed under N_2_ gas. The residues were then resuspended in MeOH and applied onto an LC-ESI-MS/MS system. The activities of both SMases were expressed as picomoles (C12-ceramide) per milligram of protein per minute.

### 2.7. Measurement of Ceramide

To assess the levels of brain tissue ceramide, followed by the extraction of ceramide, as we reported previously [36,37], the extracted lipids were dried using a vacuum system (Vision, Seoul, Korea) and then re-dissolved in methanol and analyzed by LC-ESI-MS/MS (API 5500 QTRAP mass, AB/SCIEX, Framingham, MA, USA) using a selective ion monitoring mode. The ceramide MS/MS transitions (*m*/*z*) were 510 → 264 for C14-ceramide, 538 → 264 for C16-ceramide, 566 → 264 for C18-ceramide, 594 → 264 for C20-ceramide, 648 → 264 for C24:1-ceramide, 650 → 264 for C24-ceramide, 676 → 264 for C26:1-ceramide, and 678 → 264 for C26-ceramide. Data were acquired using Analyst 1.7.1 software (Applied Biosystems, Foster City, CA, USA). Ceramide levels were expressed as picomoles per milligram of protein.

### 2.8. Detection of Neuronal Death

To confirm neuronal death, brain tissue was placed on gelatin-coated slides after cutting to 30 µm (Fisher Scientific, Pittsburgh, PA, USA). We performed Fluoro-Jade B (FJB) staining, as used by Hopkins and Schmued [38]. First, the slides were soaked in alcohol solution (100 for 3 min → 70% for 1 min) and washed in distilled water for 1 min. After that, the slides were soaked in 0.06% potassium permanganate for 15 min. Secondly, the slides were immersed in 0.001% Fluoro-Jade B (Histo-Chem Inc., Jefferson AR, USA) for 30 min and washed 3 times for 10 min each in distilled water. After drying the slide, we soaked it in xylene for 2 min and covered the top of the slide with a cover glass using DPX mounting solution (Sigma-Aldrich, Munich, Germany). We observed the fluorescence signal (450–490 nm blue excitation light) under an epifluorescence microscope. In order to quantify the result, we chose 5 coronal brain sections that were collected from each animal by cutting them into 75 µm intervals from 3.48 to 5.52 mm behind the bregma. A blinded observer counted the total number of FJB (+) cells in the hippocampal subiculum (900 × 1200 μm), CA1 (900 × 400 μm), and dentate gyrus (900 × 1200 μm) from both hemispheres under the same microscope (magnification = 10×). Data were presented as the mean number of degenerating neurons per area.

### 2.9. Immunofluorescence Analysis

To conduct immunofluorescence staining, we immersed the brain sections in 1.2% hydrogen peroxide for 20 min at RT in order to block endogenous peroxidase activity. After washing in PBS 3 times for 10 min, the primary antibodies used in this study were as follows: rabbit anti-ASMase (diluted 1:100; Invitrogen, Carlsbad, CA, USA), mouse anti-ceramide (diluted 1:10; Enzo Biochem, Inc., Farmingdale, NY, USA), rabbit anti-4HNE (diluted 1:500; Alpha Diagnostic Intl. Inc., San Antonio, TX, USA), mouse anti-CD11b (diluted 1:500; Bio-Rad, Berkeley, CA, USA), goat anti-GFAP (diluted 1:100; Abcam, Cambridge, UK), and rabbit anti-cleaved caspase-3 (diluted 1:250: Cell Signaling Technology, Danvers, MA, USA). After incubation with the primary antibodies, the brain was immersed with fluorescent-conjugated secondary antibody for 2 h at RT in a shaker (diluted 1:250 IgG and IgM (ceramide); Invitrogen, Carlsbad, CA, USA). We stained the brain sections with 4,6-diamidino-2-phenylindole (DAPI; diluted 1:1000; Invitrogen, Carlsbad, CA, USA). We placed the stained samples on gelatin-coated slides and covered them with cover slips using DPX mounting solution (Sigma-Aldrich, Munich, Germany). For ASMase, Ceramide, 4HNE, and GFAP, to quantify the intensity, the image was loaded into ImageJ v.1.47c. Then, we selected the following menu options: select image → adjust → color threshold, and darken background. Subsequently, the brightness was adjusted in accordance with the intensity of the target, and then we selected analyze → measure. To estimate the intensity of the area, we selected the menu options: select → analyze → measure. We represent the mean gray value in our data. For CD11b staining, we applied widely used functional standards of microglial activation. We used the following 3 score metric to analyze the CD11b activation: (A) Cell number score of 0: no cells are present; 1:1–9 cells; 2: 10–20 cells; 3: >20 cells within a defined area (900 × 1200 μm). (B) An intensity score of 0: no expression; 1: weak expression; 2: average expression; 3: intense expression. (C) Morphology score of 0: no activated morphology (amoeboid morphology); 1: 1–45%; 2: 45–90%; 3: >90% of microglia with activated morphology. The total score was the sum of 3 scores (cell number, intensity, and microglial morphology), ranging from 0 to 9 (magnification = 20×) [39,40]. For caspase-3, we blind counted the total number of caspase-3 (+) cells in the hippocampal CA1 (900 × 1200 μm) region (magnification = 10×).

### 2.10. Immunohistochemistry

The preconditioning process before primary antibody attachment was described in the immunofluorescence analysis above. After preconditioning, the brain sections were incubated with anti-mouse IgG secondary antibody (diluted 1:250; Vector Laboratories, Burlingame, CA, USA) in PBS containing 0.3% Triton X-100 for 2 h at RT. After washing with PBS, the brain sections were placed in ABC solution (Vector Laboratories, Burlingame, Vector, CA, USA) for 2 h, washed, and immunized with 3,3′-diaminobenzidine solution (0.06% DAB agar, Sigma-Aldrich Co., St Louis, MO, USA) in 0.01 M PBS (100 mL) containing 30% H_2_O_2_ (50 µL) for 3 min. We analyzed the immunoreactions by mounting the brain sections on gelatin-coated slides and observing the IgG leakage in the hippocampus. To analyze the IgG leakage, we proceeded with quantification in the following order of steps: (1) The image was loaded into Image J (select the menu options: image → type → 8bits, then edit → invert). (2) To measure IgG leakage, we selected the menu options analyze → measure, and the region was recorded. We represent the mean gray value in our analysis.

### 2.11. Behavioral Testing

#### 2.11.1. Assessment of Neurologic Deficits

To test whether imipramine treatment attenuated hypoglycemia-induced neurologic deficits, a modified neurological severity score (mNSS) was conducted as described previously [41]. These tests were performed at 1, 4, 7, and 14 days after hypoglycemia or sham surgery. We used the following 18-point score (0 = normal function: 18 = maximal deficit) metric to evaluate motor, sensory, balance, and reflex [42]: (A) motor tests—raising the rat by the tail (3 points) and placing the rat on the floor (3 points), (B) sensory tests (2 points), (C) beam balance tests (6 points), and (D) reflexes absence and abnormal movements (4 points). If an animal failed to perform a particular task or lacked the reflex required for the test, one point was scored. Thus, a high score means serious neurological deficits.

#### 2.11.2. Adhesive Removal Test

Starting the day after hypoglycemia, adhesive removal tests (ART) were performed for 1 week to estimate whether the administration of imipramine improved the cognitive function after hypoglycemia. After allowing the animal to adapt to the experimental conditions in a transparent test box (45 × 35 × 20 cm), we attached a sticker (1 × 1 cm) to the paws of the rat, placed the rat in the test box, and measured the time at which the rat recognized and removed the sticker. We performed 5 trials for each rat and gave each rat a minute break between trials. Each trial was assigned a maximum time of 120 s, and we recorded the maximum time if the rat could not remove the sticker within 120 s.

#### 2.11.3. Morris Water Maze Test

To test whether imipramine treatment protected against hypoglycemia-induced cognitive impairment, we conducted the Morris water maze (MWM) test for 5 consecutive days starting 9 days after hypoglycemia. For the water maze test, the equipment comprised a black circular pool (1.2 m diameter) with a hidden platform (9.5 cm diameter) submerged 1 cm below the water’s surface. The pool was filled to a depth of 30 cm with water (22–26 °C). The pool was divided into 4 equal quadrants, and an escape platform was placed in the center of zone 1 of the pool quadrants. Rats were given a place navigation test for 5 consecutive days and were dropped off at 4 different starting zones for each trial [43]. A trial began the moment the animal was placed in the water facing the wall in one of the starting zones, and the escape latency was measured at the end of the trial. After subjects located the platform and climbed up, they were relocated into a cage and waited for the next trial. For each trial, 4 trials each, and 120 s maximum, we calculated time spent and distance to the target. We also tracked the swimming route and trajectory using a camera and the SMART video tracking software 3.0 (Panlab, Carrer de l’Energia, Spain).

### 2.12. Statistical Analysis

Data were displayed as the mean ± S.E.M. Statistical significance between experimental groups was measured by analysis of variance (ANOVA) in accordance with the Bonferroni post hoc test. Differences were considered statistically significant at *p* < 0.05.

## 3. Results

### 3.1. Imipramine Reduced Hypoglycemia-Induced ASMase Activation and Ceramide Production

ASMase is excessively activated in pathological states, and the activated ASMase catalyzes the decomposition of sphingomyelin into ceramide [44,45,46,47,48]. First, we performed ASMase and ceramide staining to establish whether hypoglycemic injury activates ASMase and thereby increases the generation of ceramide in the hippocampus. As Figure 1 shows, ASMase activation and ceramide production were rarely observed in the sham-operated groups, while the hypoglycemia-induced group showed a significant increase in ASMase activation and ceramide production. However, in the group administered imipramine for 7 days after hypoglycemia, ASMase activation and ceramide production were greatly reduced (Figure 1A). In addition to this histological analysis, we also measured changes in ASMase and NSMase activity and analyzed the concentration of 11 major ceramide sub-species that are commonly observed in brain neurological disorders [49,50], such as within the hippocampus after hypoglycemia. With the SMase activity measurements, we confirmed that ASMase existed at a higher basal concentration than NSMase in the sham-vehicle group (Figure 1C). Figure 1C shows that the hypoglycemia-induced group had increased ASMase and NSMase activity compared to the sham-operated groups. However, imipramine treatment only reduced the levels of ASMase activity following hypoglycemia. Furthermore, an analysis of 11 types of ceramide derivatives performed by measuring the N-acetyl chain length showed a significant increase in the total amount of ceramide in the hypoglycemia-induced group compared to the sham-operated groups as a whole (Appendix A). In particular, C18 and C24:1 were present at higher concentrations in the sham-operated groups than the other ceramide types. We believe that both types may be the most common species of ceramide in the hippocampus. This showed a significant change in the hypoglycemia-induced group, which had a greatly increased C18 and C24:1 concentration over the baseline, while imipramine decreased both types of ceramides to a similar level as seen in the sham-operated groups (Figure 1D). These results confirm that hypoglycemic injury leads to an increase in ASMase and a subsequent increase in ceramide, which imipramine inhibits.

### 3.2. Imipramine Reduced Hypoglycemia-Induced Neuronal Death

Severe hippocampal neuronal death is not caused by a single event, but rather it is sequentially caused by the primary introduction of the hypoglycemic state, followed by a secondary injury that is dependent on glucose reperfusion used to clinically recover the hypoglycemic condition. Severe hypoglycemia usually results in neuronal death in the subiculum, CA1, and dentate gyrus (DG) regions of the hippocampus [8]. We administered imipramine (10 mg/kg, i.p.) for seven days after hypoglycemia and sacrificed animals on the seventh day. FJB staining is a selective marker for degenerating neurons. Then, we performed FJB staining to determine how imipramine affects neuronal death caused by hypoglycemia. We verified that hypoglycemia severely increased degenerating neurons in the subiculum, CA1, and DG regions of the hippocampus, while administration of imipramine significantly reduced neuronal death after hypoglycemia (Figure 2).

### 3.3. Imipramine Reduced Hypoglycemia-Induced Oxidative Injury

Severe hypoglycemia causes NADPH oxidase to become excessively activated, producing reactive oxygen species (ROS) that cause oxidative damage to the brain. Moreover, 4-hydroxynoneal (4-HNE) is produced by lipid peroxidation in cells and is generally used as a marker of oxidative stress under pathological conditions. Thus, we conducted 4-HNE staining to confirm the effects of imipramine on the oxidative injury induced by hypoglycemia. For the sham-vehicle and sham-IMP groups in Figure 3, we observed little oxidative stress. However, in the hypoglycemia-vehicle group, insult caused significant oxidative stress in subi, CA1, and DG in the hippocampus. In the group administered imipramine for 7 days following hypoglycemia, oxidative stress was greatly reduced to approximately the levels of the sham-operated groups in all three regions of the hippocampus (i.e., subi, CA1, and DG). As a result, this study confirmed that imipramine is also effective in inhibiting hypoglycemia-induced oxidative damage (Figure 3).

### 3.4. Imipramine Reduced Hypoglycemia-Induced Microglia and Astrocyte Activation

The excessive activation of microglia and astrocytes in brain diseases is known to promote inflammatory processes, and these reactions are known to cause secondary injury in the brain. To verify the activation of microglia and astrocytes, we conducted CD11b and GFAP staining, respectively, and observed only a small number of microglia and astrocytes in the rest of the sham-operated groups. In the severe hypoglycemia model used in this study, hypoglycemic insult caused the excessive activation of microglia and astrocytes in the hippocampus, particularly in the CA1 region, which is regarded as one of the most vulnerable sites to neuronal death [9,10]. On the other hand, this study found significantly reduced activation of microglia and astrocytes in the group administered with imipramine after hypoglycemia (Figure 4).

### 3.5. Imipramine Prevented Hypoglycemia-Induced Blood–Brain Barrier (BBB) Disruption

To clarify whether imipramine has a protective effect on BBB disruption caused by hypoglycemia, we measured the degree of lgG leakage. As a result of the IgG staining, the sham-operated groups did not have lgG leakage, meaning that the tissue did not react chemically. Thus, we observed that the color of the tissue was transparent. On the other hand, when hypoglycemia occurred, IgG leakage from damaged blood vessels caused a chemical reaction with the dye, oxidizing it and turning the tissue darker, resulting in a gradation of color in the tissue that was proportional to the degree of leakage. Figure 5 confirms that IgG leakage was severe in the hypoglycemia group, while this leakage decreased in the imipramine-treated group following hypoglycemia. This suggests that imipramine is effective in preventing BBB disruption (Figure 5).

### 3.6. Imipramine Reduced Apoptosis-Associated Hypoglycemia-Induced Cleaved Caspase-3

Cleaved caspase-3 is an important molecule involved in pro-apoptotic signaling and neural apoptotic pathways. [51,52]. The species of ceramide that we focused on in this study is also known to induce cleaved caspase-3 [53,54,55]. Therefore, we conducted cleaved caspase-3 staining to verify that the cleaved caspase-3 expression in the hippocampus following hypoglycemia was consistent with the changes we observed in the ceramide levels. As the results of Figure 6 show, cleaved caspase-3 increased significantly in the hypoglycemia group compared to the sham-operated groups. However, following injury in the imipramine-treated group, the concentration of cleaved caspase-3 was reduced significantly. These results indirectly suggest that the administration of imipramine, a known ASMase inhibitor, reduced the generation of ceramide, thereby decreasing the amount of ceramide-induced cleaved caspase-3 and consequently inhibiting apoptosis (Figure 6).

### 3.7. Imipramine Improved Neurologic and Cognitive Function after Hypoglycemia

To functionally measure neurologic and cognitive dysfunction, we performed a modified neurological severity score (mNSS), a standard adhesive removal test (ART), and the Morris water maze (MWM) after the induction of hypoglycemia. These tests particularly assess motor-, sensory-, and memory-related brain regions such as the cortex and hippocampus after hypoglycemia. First, we performed an mNSS test on days 1, 4, 7, and 14 to confirm whether imipramine treatment could reduce hypoglycemia-induced neurological dysfunction. As a result, the hypoglycemia-imipramine group received a lower mNSS score than the hypoglycemia-vehicle group, which means a decrease in neurological dysfunction (Figure 7A). We also found that imipramine treatment improved cognitive ability via ART and MWM. For ART, we checked the cognitive function for 7 days, treating with imipramine. After placing the test subjects in a transparent box with a 1 × 1 cm sticker on the rats’ forepaws, we observed the rats’ behavior and determined whether the rat could recognize and detach the tape within 120 s. The sham-operated groups recognized the tape as soon as they entered the box, waving their forepaws, and took less than 10 s to remove the tape. In the case of the hypoglycemia-vehicle group, most rats were unaware of the tape attached to their front paws and took more than one minute to remove it. However, the imipramine-treated hypoglycemia group removed the tape increasingly rapidly over a six-day period. The hypoglycemia-imipramine group took 50 s to take off the tape on the first day of the test and detached the tape at a similar rate to the sham-operated groups on the last day (Figure 7B). Since the administration of imipramine was highly effective for a week, we considered it meaningless to continue it for more than 7 days and investigated how hypoglycemia-induced cognitive dysfunction changes after administration of imipramine. For MWM, we conducted it for 5 consecutive days, starting 9 days after hypoglycemia. As a result, we verified that there was no difference between the imipramine-treated and the vehicle-treated groups on the first day. However, over time, the imipramine-treated group quickly reached the target platform, and the distance to the target was short compared to the vehicle-treated group (Figure 7C–E).

## 4. Discussion

In this study, we conducted various staining methods using brain tissue, in addition to performing a cognitive impairment analysis to test whether the administration of imipramine provides neuronal protection against severe hypoglycemia. This work is important in determining whether ASMase and ceramide have an effect on neuronal death and whether imipramine can prevent neuronal death and cognitive dysfunction.

Our research team found that when severe hypoglycemia occurs, the vesicular zinc in the pre-synapse is excessively secreted into the synaptic cleft, which causes excessive zinc to be absorbed and accumulated into the post-synapse, leading to hippocampal neuronal death [12]. According to previous studies, zinc increases the activation of ASMase, which contains severely highly conserved zinc-binding motifs and allows the activated ASMase to convert sphingomyelin to ceramide, resulting in excessive ceramide generation [56,57,58,59]. Thus, we hypothesized that zinc may increase the production of ASMase in hypoglycemia (Figure 8).

Although ASMase plays an important role in maintaining the cell membrane under normal conditions, excessive ASMase activation promotes abnormal ceramide production under pathological conditions [60,61,62,63]. This condition triggers autophagy, high lysosomal membrane permeability, the release of immune cytokines, apoptosis, necrosis, etc. [44,46,64,65,66,67]. In this study, we know that ASMase was abnormally increased after injury, and we checked that excessive ceramide was formed in the hippocampus. This means that zinc was excessively released due to the fact of hypoglycemia and activated ASMase to promote the production of ceramide. Imipramine only reduced ASMase activity and did not affect NSMase, meaning that imipramine specifically inhibited ASMase. In addition, C18 and C24:1 existed at high concentrations even in normal conditions, and these two types increased significantly after injury, indicating indirectly that C18 and C24:1 are involved in the hypoglycemia-induced neuronal death pathway. It is unclear how the decrease of C18 and C24:1 can be related to neuroprotection. However, in this study, we suggest that changes in the concentrations of C18 and C24:1 play an important role in hypoglycemia-induced neuronal death and that C18 and C24:1 may be important targets for inhibiting neuronal death after hypoglycemia.

Abnormal redox signaling is involved in diverse pathophysiological conditions such as inflammation, hypoxia, ischemia/reperfusion, diabetes mellitus, and other neurodegenerative diseases [68,69,70,71,72]. As with the abovementioned neurological disorders, in the case of oxidative stress after hypoglycemia, our research team found that zinc acts on p47 in the cytoplasm and produces ROS by attaching p47 to NADPH oxidase in the cell membrane [73,74]. A previous study suggests that the formation of ROS is an event downstream of ASMase activation [16]. In addition, several studies have shown that ceramide acts on the mitochondria to form ROS [75,76,77]. This study revealed that the administration of imipramine may be used as a way to inhibit neuronal death by reducing ceramide-induced ROS among hypoglycemia/glucose reperfusion-induced ROS production factors.

According to some papers, diverse diseases have shown that ceramide causes abnormal immune responses and leads to secondary injury in the tissues [78,79]. Ceramide is part of pro-inflammation signaling and promotes inflammation through various pathways. Ceramide activates the Nlrp3 inflammasome to produce active pro-inflammatory IL-1β or TNFα and stimulates Ca^2^⁺-dependent cytosolic phospholipase A2 (cPLA2), causing Cox2 to lead to inflammation and pain [80,81,82]. Under pathological conditions, excessive neuroinflammation can deal with damaged cells but also attack normal healthy cells nearby, thus enhancing neuronal death. In the pathological state of the brain, the excessive activation of glial cells induced an immune response, and after severe hypoglycemia, these glial cells were excessively activated compared to those in the sham-operated groups. However, the group treated with imipramine after hypoglycemia showed a decrease in glial cell activation, such as in microglia and astrocytes via inhibiting ceramide formation. This indicates that, in hypoglycemia, the ASMase/ceramide pathway is involved in extensive inflammation in the brain, which can lead to greater neuronal cell death.

In the brain, the blood–brain barrier (BBB) plays a major role in the brain’s exchange of materials and in maintaining brain homeostasis. When extreme hypoglycemia occurs, ROS production and seizures occur and the physiological balance is upset as soon as iso-EEG occurs; following this, BBB destruction begins [9,10]. Although ASMase is expressed in all cell types under normal conditions, previous studies have shown that the activation of ASMase causes age-related brain damage and is linked to the destruction of the BBB, or that ASMase deficiency inhibits BBB leakage and tight junction disruption [15,45]. Other studies have shown that ceramide is increased at the end of the astrocytes supporting the brain vasculature, which directly acts on BBB function via increasing BBB loss and BBB permeability [83,84]. Therefore, ASMase/ceramide system control is an important means to reduce BBB disruption after hypoglycemia. In addition, moderate ROS generation destroys the tight junction, resulting in BBB disruption [83,85]. This implies that BBB disruption can occur due to the fact of ceramide-induced ROS as well as the ASMase/ceramide system itself. However, the link between ASMase, ceramide, and BBB disruption in hypoglycemia is unclear. Therefore, in this study, we investigated the effect of suppressing BBB disruption by inhibiting ASMase and reducing ceramide-induced ROS formation. As a result, we discovered that the administration of imipramine following hypoglycemia inhibited BBB disruption via controlling the ASMase/ceramide system and ceramide-induced ROS.

Emery and Clark revealed that cleaved caspase-3 resulted in retinal cell apoptosis and neuronal death under conditions of severe hypoglycemia and traumatic brain injury, respectively [52,86]. Additionally, other studies have found that ceramide activates cleaved caspase-3, and then neuronal cell apoptosis occurs through the caspase pathway [53,54,55]. This study identified the relationship between cleaved caspase-3 and hypoglycemia-induced neuronal apoptosis in the hippocampus, as well as whether the administration of imipramine after hypoglycemia could decrease ceramide-induced cleaved caspase-3. This study suggests that ceramide-induced cleaved caspase-3 after hypoglycemia can be suppressed by administering imipramine, which consequently inhibits neuronal apoptosis.

Finally, we performed the modified neurological severity score (mNSS), the adhesive removal test (ART), and the Morris water maze (MWM) to identify hypoglycemia-induced neurological and cognitive impairment and to determine whether the administration of imipramine could improve the outcome. In this study, we verified that imipramine significantly improved the neurological and cognitive impairment caused by hypoglycemia. Of course, we believe that detailed studies are needed on how ceramide, especially C18 and C24:1, causes hippocampal neuronal death after hypoglycemia in the future. This study provides background data for basic and clinical research in studying the mechanisms of ceramide-induced hippocampal neuronal death after hypoglycemia. Therefore, this study suggests that the administration of imipramine when severe hypoglycemia occurs in diabetic patients prescribed insulin could be a potentially promising therapeutic approach to prevent hippocampal neuronal death and cognitive deficit during hypoglycemia/glucose reperfusion.

## 5. Conclusions

In the present study, we researched whether imipramine, an ASMase inhibitor, had neuroprotective effects on hypoglycemia-induced neuronal death. We confirmed that an increase in ASMase activation and ceramide formation led to hippocampal neuronal death after hypoglycemia. Our present study proposes that imipramine has strongly neuroprotective effects. Thus, we suggest that the inhibition of ASMase could be a therapeutic approach to prevent neuronal death after severe hypoglycemia.

## Figures and Tables

**Figure 1 cells-11-00667-f001:**
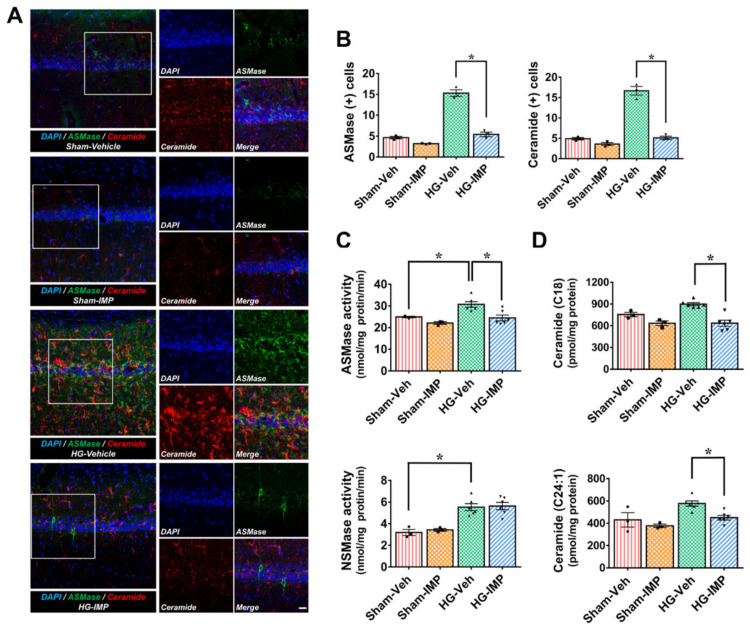
Effects of imipramine (IMP) on hypoglycemia-induced acidic sphingomyelinase (ASMase) activation and ceramide production. Fluorescent images show the effect of imipramine treatment on ASMase activation and ceramide production after hypoglycemia: (**A**) difference in ASMase (green) and ceramide (red) intensity between vehicle- and IMP-treated groups in the vulnerable CA1 region following hypoglycemia. Under normal conditions, neuronal cells maintained an adequate amount ASMase and ceramide, while hypoglycemia caused an excessive rise in both. Scale bar = 10 µm. (**B**) Bar graph shows the quantification of ASMase and ceramide intensity in the CA1 region. (**C**) The difference in ASMase and NSMase activity between vehicle- and IMP-treated groups in the hippocampus following hypoglycemia. (**D**) An analysis of the ceramide content (C18 and C24:1) in the hippocampus. Compared to the sham-operated groups, ceramide contents were found to be increased in the hypoglycemia condition and significantly reduced in ceramide contents when imipramine was administered. Data are the mean ± SEM; *n* = 3 from each sham group, *n* = 6 from each hypoglycemia group. Individual data points (● = Sham-Veh, ■ = Sham-IMP, ▲ = HG-Veh, ▼ = HG-IMP). * Significantly different from the vehicle-treated group at *p* < 0.05.

**Figure 2 cells-11-00667-f002:**
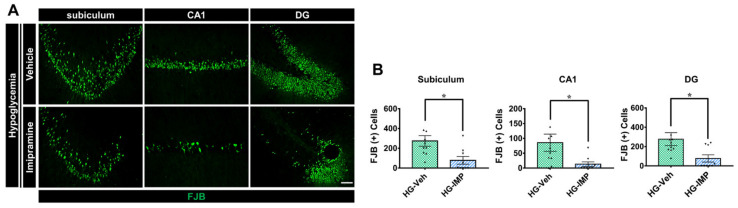
Effects of imipramine (IMP) on hypoglycemia-induced hippocampal neuronal death. Hypoglycemia resulted in hippocampal neuronal death in the subiculum (subi), CA1, and DG regions. (**A**) Fluoro-Jade B (FJB)-positive neuronal cells, degenerating cells, were observed in the subi, CA1, and DG regions after hypoglycemia. Scale bar = 50 µm. (**B**) The number of FJB (+) cells was greatly lower in the imipramine-treated group than in the vehicle group after insult. Data are the mean ± SEM; *n* = 9–10 from each hypoglycemia group. Individual data points (▲ = HG-Veh, ▼ = HG-IMP). * Significantly different from the vehicle-treated group at *p* < 0.05.

**Figure 3 cells-11-00667-f003:**
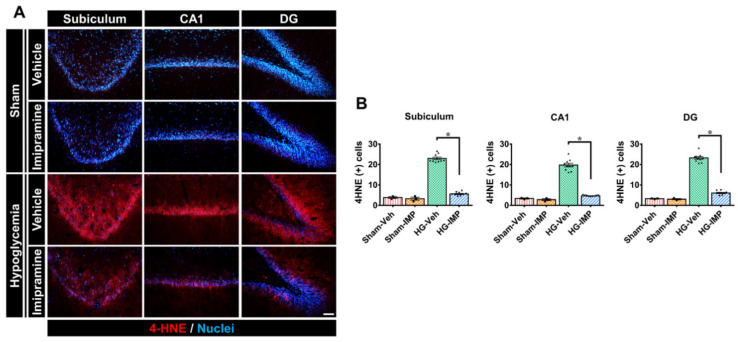
Administration of imipramine reduced oxidative stress after hypoglycemia. Neuronal oxidative stress was detected by 4-hydroxy-2-nonenal (4HNE) staining in the hippocampus: (**A**) sham-operated groups rarely presented the 4HNE signal in the hippocampus. While hypoglycemia caused oxidative stress significantly, imipramine decreased oxidative stress in all three regions of the hippocampus. Scale bar = 50 µm. (**B**) Quantification of 4HNE fluorescence intensity in the hippocampus. The fluorescence indicates a significant gap between the vehicle- and imipramine-treated groups. Data are the mean ± SEM; *n* = 3 for each sham group, *n* = 9–10 for each hypoglycemia group. Individual data points (● = Sham-Veh, ■ = Sham-IMP, ▲ = HG-Veh, ▼ = HG-IMP). * Significantly different from the vehicle-treated group at *p* < 0.05.

**Figure 4 cells-11-00667-f004:**
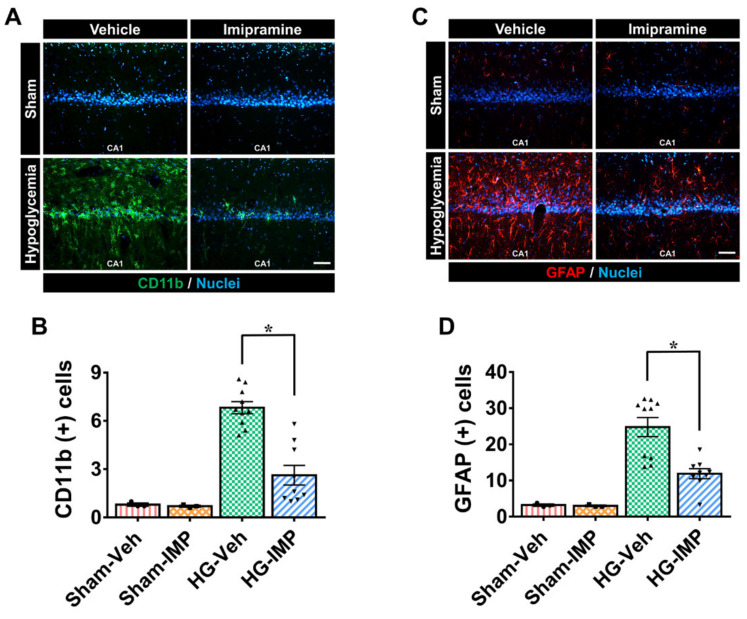
Administration of imipramine reduced microglial and astrocyte activation after hypoglycemia. To determine microglia and astrocyte activation, we conducted CD11b and GFAP staining, respectively. Hypoglycemia-induced microglia and astrocyte activation trigger an immune response. CA1 is vulnerable to injury by activated glial cells. (**A**) The green fluorescence (CD11b) shows activated microglia in the CA 1 region. Sham-operated groups displayed almost no microglial activation signal. However, after hypoglycemia, microglia cell intensity, number, and morphology were greatly boosted in the vehicle-treated group compared to the imipramine-treated group. (**B**) The grade of microglia activation in the CA1. (**C**) The red fluorescence (GFAP) shows activated astrocytes in the CA1 region. Sham-operated groups displayed almost no astrocyte activation, while the hypoglycemia-vehicle group showed significantly increased fluorescence intensity. (**D**) The bar graph represents the intensity of activated astrocytes in the CA1. However, after injury, imipramine prevented microglia and astrocyte activation in the CA1. Scale bar = 50 µm. Data are the mean ± SEM; *n* = 3 for each sham group; *n* = 9–10 for each hypoglycemia group. Individual data points (● = Sham-Veh, ■ = Sham-IMP, ▲ = HG-Veh, ▼ = HG-IMP). * Significantly different from the vehicle-treated group at *p* < 0.05.

**Figure 5 cells-11-00667-f005:**
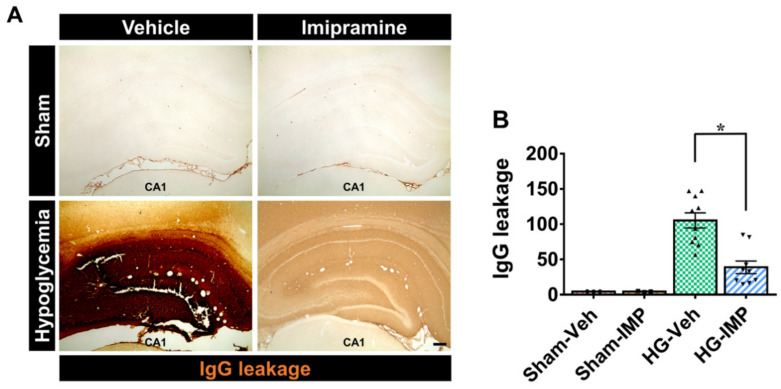
Administration of imipramine reduced blood–brain barrier (BBB) disruption after hypoglycemia. This figure shows leakage of IgG serum through BBB disruption in the hippocampus after hypoglycemia: (**A**) A 4× magnification of a microscopic image of IgG in the hippocampus in each group. These images display that BBB breakdown occurred after hypoglycemia. The hypoglycemia-imipramine-treated group showed a decrease in the leakage of IgG serum compared with the hypoglycemia-vehicle group. Scale bar = 200 µm. (**B**) The bar graph represents the quantification of IgG serum extravasation in the hippocampus. Data are the mean ± SEM; *n* = 3 for each sham group, *n* = 9–10 for each hypoglycemia group. Individual data points (● = Sham-Veh, ■ = Sham-IMP, ▲= HG-Veh, ▼ = HG-IMP). * Significantly different from the vehicle-treated group at *p* < 0.05.

**Figure 6 cells-11-00667-f006:**
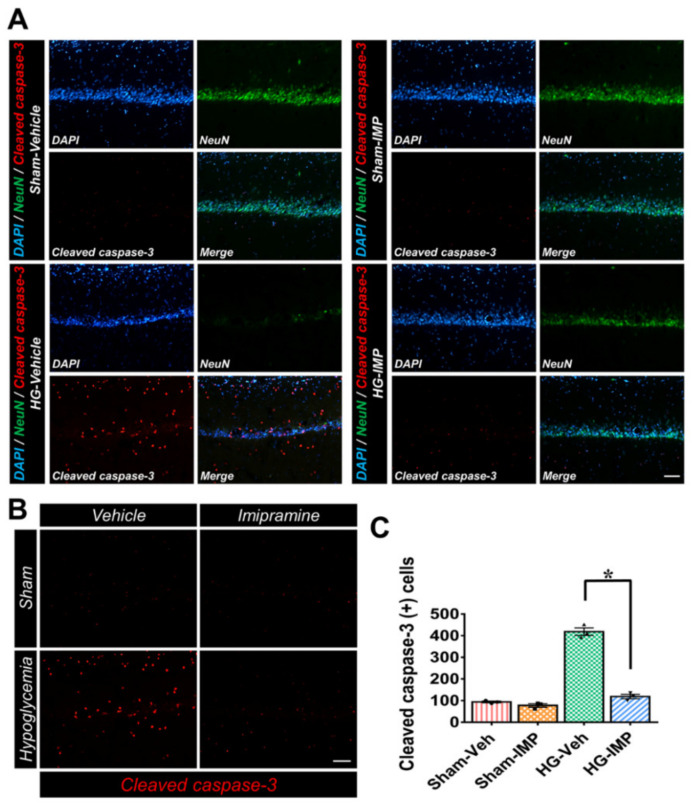
Administration of imipramine reduced the number of apoptotic cells after hypoglycemia: (**A**) cleaved caspase-3 (red) and NeuN (green) in the hippocampus CA1 region. After hypoglycemia, the administration of imipramine for 1 week decreased the number of apoptotic cells and increased the number of live neurons compared to the hypoglycemia-vehicle group. Scale bar = 100 µm. (**B**) The bar graph indicates the quantification of the apoptotic cell number. Data are the mean ± SEM; *n* = 3 from each group. Individual data points (● = Sham-Veh, ■ = Sham-IMP, ▲ = HG-Veh, ▼ = HG-IMP). * Significantly different from the vehicle-treated group at *p* < 0.05.

**Figure 7 cells-11-00667-f007:**
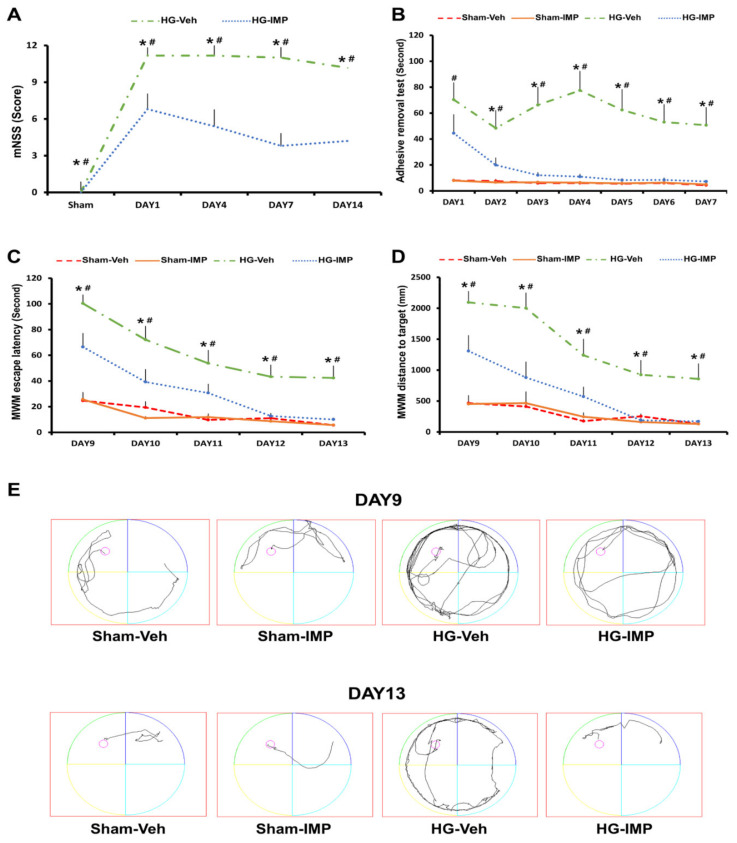
Administration of imipramine improved neurologic and cognitive function after hypoglycemia: (**A**) the mNSS was scored in rats at 1, 4, 7, and 14 days after hypoglycemia. A score of 18 indicated that all trials had failed; a score of 0 indicated that all trials were completed. (**B**) ART performance test for 1 week: the imipramine-treated group removed the tape faster than the vehicle-treated group after hypoglycemia. (**C**–**E**) The MWM performance test. (**C**) The target platform arrival time for 5 consecutive days starting post-injury day (PID) 9. (**D**) The distance to target: the imipramine-treated group had a shorter distance to reach the target over time than the vehicle-treated group. (**E**) The average tracking record of each group reaching the target. Data are the mean ± SEM; *n* = 3 for each sham group. *n* = 5–7 for each hypoglycemia group. * Significantly different from the vehicle-treated group at *p* < 0.05, # significantly different from the sham-operated group at *p* < 0.05.

**Figure 8 cells-11-00667-f008:**
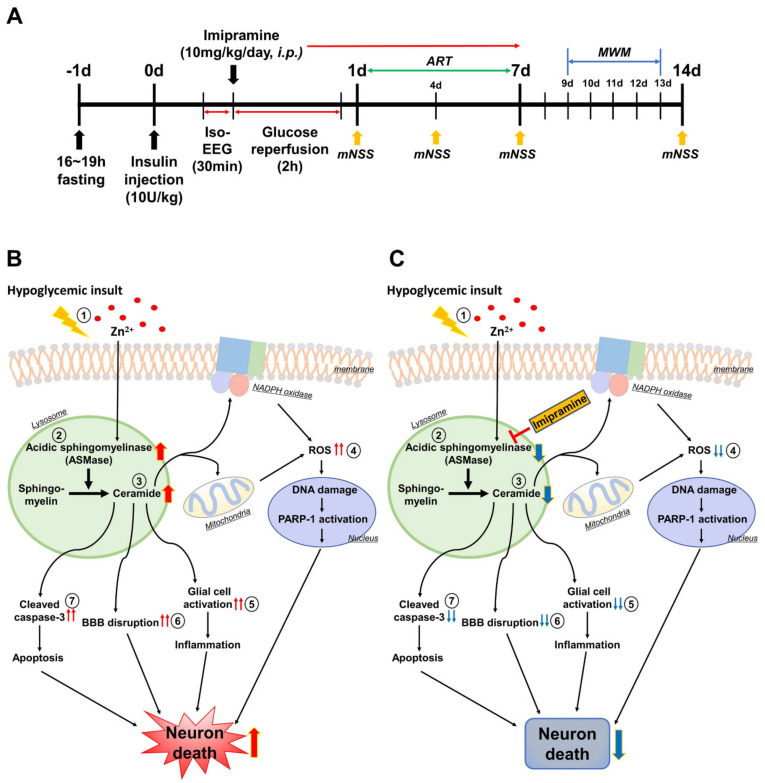
Possible association of ASMase, ceramide, and neuronal death under hypoglycemia/glucose reperfusion conditions. This schematic illustration explains the effects of imipramine on the process of the hypoglycemia-induced neuronal death mechanism. (**A**) Experimental timeline. (**B**) Hypoglycemia-induced neuronal death mechanism: (**1**) after hypoglycemia, excessive zinc release and translocation from pre-synapse to post-synapse; (**2**) the excessive zinc activates ASMase abnormally; (**3**) the activated ASMase decomposes sphingomyelin into ceramide; (**4**) the increased ceramide acts on the mitochondria and NADPH oxidase, increasing ROS production; (**5**) the increased ceramide activates glial cells, causing inflammation; (**6**) the increased ceramide causes blood–brain barrier disruption; (**7**) the increased ceramide increases capase-3, resulting in apoptosis. When these conditions dominate, neuronal death is more likely to occur severely. (**C**) The effects of imipramine on hypoglycemia-induced neuronal death. The administration of imipramine can inhibit excessive ASMase activation and then ceramide production. Therefore, there is a possibility that imipramine can prevent neuronal death after hypoglycemia.

## Data Availability

Not applicable.

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
