# Peer review of "Administration of an Acidic Sphingomyelinase (ASMase) Inhibitor, Imipramine, Reduces Hypoglycemia-Induced Hippocampal Neuronal Death"

_cells, 2022, doi:10.3390/cells11040667_

Round 1

Reviewer 1 Report

The manuscript  ´Administration of an acidic sphingomyelinase (ASMase) inhibitor, imipramine, reduces hypoglycemia-induced hippocampal 3 neuronal death´ submitted by Kho et al. indicates in an animal model that acidic sphingomyelinase (ASM) is activated in response to hypoglycemia in the hippocampus. This is accompanied by neuronal stress and cell death. The use of imipramine, an ASMase inhibitor, reduces the neurological dysfunction.

The authors performed a hypoglycemic animal experiment and treated one group with imipramine. Indeed, the glycemic neurological dysfunctions were reduced in the presence of imipramine. The authors did a variety of experiments to indicate that hypoglycemia is connected with an activation of ASMase. Inhibition of ASMase with imipramine reduced neurological changes. Although the study is well performed, the conclusion is not fully supported. Thus, the study has 2 weak points. First, it is not certain that the hypoglycemic-induced dysfunctions are due to activation of ASM. This could of course be demonstrated in ASM knockout animals that exist. The 2nd point is that imipramine is not a specific ASM inhibitor. As an antidepressive drug, neurotransmitters are increased which may also contribute to an effect. Indeed, the protective effect of imipramine is very well known, but other mechanisms (epigenetics, neurotransmitters) are also discussed. Especially, the cognitive impairment experiments could be a result of neurotransmitter function.  Thus, mirtazepine also has neuroprotective effects, although it does not inhibit ASM. These two experiments would have had to be performed to show clear relationships, but this will be too time-consuming. Thus, the authors should discuss this adequately.

The following points should be addressed:

The authors stated that the rats were divided into 4 groups to determine the effects of imipramine. The animals were given an intraperitoneal injection of IMP (10 mg/kg, i.p.) once per day for 7 days, then the animals were sacrificed. What animals were used for the cognitive function as the experiments were done after 9 – 14 days after glycemia. Was this an additional animal experiment and received the animals imipramine every day? Nothing is stated in the manuscript.

The authors measured 11 different ceramide levels and present two of them in the manuscript. Ceramide levels are presented as pmol/mg protein. It would be important to measure the corresponding sphingomyelines to see whether there are also changes due to ASM activity.

In Figure 2 the effects of imipramine on hypoglycemia-induced hippocampal neuronal death is shown. Hypoglycemia resulted in hippocam-251 pal neuronal death in the subiculum, CA1, and DG regions but no negative control is shown. Are there no FJB positive cells visible?

The authors performed the modified neurological severity score, the adhesive removal test and the Morris water maze to identify hypoglycemia-induced neurological and cognitive impairment. The administration of imipramine improved the neurological and cognitive impairment caused by hypoglycemia. However, these results seem to be isolated from the former experiments and it is likely that especially these effects are due to neurotransmitter changes.

Author Response

Reviewer 1

The manuscript ´Administration of an acidic sphingomyelinase (ASMase) inhibitor, imipramine, reduces hypoglycemia-induced hippocampal neuronal death´ submitted by Kho et al. indicates in an animal model that acidic sphingomyelinase (ASM) is activated in response to hypoglycemia in the hippocampus. This is accompanied by neuronal stress and cell death. The use of imipramine, an ASMase inhibitor, reduces the neurological dysfunction.

The authors performed a hypoglycemic animal experiment and treated one group with imipramine. Indeed, the glycemic neurological dysfunctions were reduced in the presence of imipramine. The authors did a variety of experiments to indicate that hypoglycemia is connected with an activation of ASMase. Inhibition of ASMase with imipramine reduced neurological changes. Although the study is well performed, the conclusion is not fully supported. Thus, the study has 2 weak points. First, it is not certain that the hypoglycemic-induced dysfunctions are due to activation of ASM. This could of course be demonstrated in ASM knockout animals that exist. The 2nd point is that imipramine is not a specific ASM inhibitor. As an antidepressive drug, neurotransmitters are increased which may also contribute to an effect. Indeed, the protective effect of imipramine is very well known, but other mechanisms (epigenetics, neurotransmitters) are also discussed. Especially, the cognitive impairment experiments could be a result of neurotransmitter function.  Thus, mirtazepine also has neuroprotective effects, although it does not inhibit ASM. These two experiments would have had to be performed to show clear relationships, but this will be too time-consuming. Thus, the authors should discuss this adequately.

The following points should be addressed:

  1. The authors stated that the rats were divided into 4 groups to determine the effects of imipramine. The animals were given an intraperitoneal injection of IMP (10 mg/kg, i.p.) once per day for 7 days, then the animals were sacrificed. What animals were used for the cognitive function as the experiments were done after 9 – 14 days after hypoglycemia. Was this an additional animal experiment and received the animal imipramine every day? Nothing is stated in the manuscript.

<Response: We appreciate this reviewer’s comment. As you mentioned, we conducted an additional animal experiment for this revision and we treated with imipramine for 7 days after hypoglycemia. We performed a Morris water maze for 5 days from 9 to 13 days and an mNSS test on days 1, 4, 7, and 14. Since the administration of IMP was highly effective for a week, we considered it meaningless to administer it for more than 7 days, and we tried to evaluate how the inhibition of hypoglycemia-induced neuronal death via the administration of IMP affected cognitive function. This content was added to Results 3.7 and Figure 7 legend>

  1. The authors measured 11 different ceramide levels and present two of them in the manuscript. Ceramide levels are presented as pmol/mg protein. It would be important to measure the corresponding sphingomyelins to see whether there are also changes due to ASM activity.

<Response: We appreciate this reviewer’s comment. That is a suitable idea. We also measured sphingomyelin, but there was little difference between the hypoglycemia-vehicle and the hypoglycemia-imipramine groups. We speculate that cells maintain a certain level of sphingomyelin bu do not know why. It would be efficacious to conduct more detailed research in the future.>

  1. In Figure 2 the effects of imipramine on hypoglycemia-induced hippocampal neuronal death is shown. Hypoglycemia resulted in hippocampal neuronal death in the subiculum, CA1, and DG regions but no negative control is shown. Are there no FJB positive cells visible?

<Response: We appreciate this reviewer’s comment. Sham-operated groups did not show any degenerating neurons. Thus, we performed quantification of FJB staining (Figure 2), except for on sham-operated groups.>

  1. The authors performed the modified neurological severity score, the adhesive removal test and the Morris water maze to identify hypoglycemia-induced neurological and cognitive impairment. The administration of imipramine improved the neurological and cognitive impairment caused by hypoglycemia. However, these results seem to be isolated from the former experiments and it is likely that especially these effects are due to neurotransmitter changes.

<Response: We appreciate this reviewer’s comment. As the reviewer mentioned, these may be effects that occur due to neurotransmitter changes. However, we think this study is meaningful in that it was the first discovery to confirm the association between hippocampal neuronal death and ASMase/ceramide signaling after hypoglycemia. Moreover, as you mentioned, it is interesting to confirm the changes in neurotransmitters between the hypoglycemia-vehicle and hypoglycemia-imipramine groups. By confirming this, research on the association between hypoglycemia-induced neuronal death and ASMase/ceramide signaling can be conducted in more depth in the future.>

Reviewer 2 Report

The purpose of the study is to test the hypothesis that imipramine could decrease hippocampal neuronal death after hypoglycemia by the inhibition of ASMase with consequent reduction of ceramide. The novelty of this study is related to the use of the ASM-inhibitor imipramine to revert hypoglycemia-induced neuronal death.

The manuscript needs some improvements in order to be ready for publication,  as described below.

It is already well known that agents that reduce ASM activity and thereby also ceramide levels tend to attenuate receptor-mediated apoptosis, stress stimuli-induced apoptosis, as well as growth factor-deprivation-mediated apoptosis and promote cell proliferation. Thus, ASM-inhibitors potentially have antiapoptotic and neuroprotective effects and have been extensively studied for the treatment of disorders such as brain ischemia, stroke, ethanol-induced neuronal cell death, Alzheimer’s dementia, Parkinson’s disease, Chorea Huntington, spinal cord injury, seizure disorder, glaucoma, and to protect against neurodegeneration occurring in multiple sclerosis. Specifically, it has been shown that therapeutic concentrations of the antidepressants amitriptyline and fluoxetine functionally inhibit Asm and reduce ceramide concentrations in the hippocampus (https://doi.org/10.1159/000442611). Curiously, none of these are mentioned in the introduction section or discussed, and it gives the wrong impression of a new idea. However, it has been well explored in the literature so far.

  1. Abstract needs to be improved. There are a lot of single sentences not connected. Avoid repeating “the present study” more than once.
  2. Review papers rather than original papers have been cited in most of the references in the Introduction and Discussion sections. Please use original articles.
  3. (Lane 25-26) – Please give some numerical data on diabetes/diabetic patients.
  4. (Line 27-29) - Please, explore the sentence, “During the last few decades, many diabetes patients have experienced hypoglycemia while attempting to tightly control their blood glucose to prevent diabetic complications.” How frequently does it happen? Or how many patients? Give some numerical data about the subject.
  5. How to correlate repeated hypoglycemia (1st paragraph) with severe hypoglycemia (2nd paragraph)?
  6. (Line 36-38) – The statement “our lab has demonstrated that glucose reinstatement can cause an increase in neuronal death after hypoglycemia. Thus, we called it glucose reperfusion injury after hypoglycemia [10]” gives the wrong impression that authors research group described this classic pattern that was tested experimentally in a rat model of ischemia/reperfusion by Pulsinelli et al., several decades ago and repeated by several research groups. Please rephrase it.
  7. (Line 43-46) - All sentences started in the same way. Please change it. The sentences need to be integrated, telling a story. And not isolated sentences, as they are. It is mentioned three times that Sphingomyelin is found/ located/ component of cell membranes. Please, rewrite.
  8. (Line 50-57) - Same as above, all sentences started in the same way, and no connection between them. Please, rewrite.
  9. Please describe the “Animal Surgery and Severe Hypoglycemia Induction” keeping the order of the events: After overnight fasting, animals were anesthetized (description)….then a catheter was inserted… and so on….
  10. (Line 93) – describe the anesthesia procedure: which anesthesia was used? How much was it used? For how long? Was there a maintenance dose?
  11. When, after hypoglycemia induction, was IMP given intraperitoneally into the animals?
  12. What does the sentence below mean? “as time goes by, the sucrose solution permeates into the brain and touches the bottom, which means that the cryosection is ready for immunostaining.” Please, describe it correctly.
  13. (Lines 247-248) – rephase the sentence “FJB showed that we found large amounts of degenerating neurons...”
  14. What it is 4-HNE (4-hydroxy-2-nonenal)? Moreover, what does it do?
  15. Grammar revision is necessary for section “3.7. Imipramine improves neurologic and cognitive function after hypoglycemia.”
  16. In Figure 7, do not repeat the text in the main manuscript. Please, correct.

The discussion section needs to be rewritten entirely. Most of the discussion section is a repetition of information already in the introduction section and repletion of results descriptions. Instead, results should be discussed based on their novelty and correlated with the literature. Figures should not be mentioned in this section either. There is no discussion at all.

  1. General English improvement.

Author Response

Reviewer 2

The purpose of the study is to test the hypothesis that imipramine could decrease hippocampal neuronal death after hypoglycemia by the inhibition of ASMase with consequent reduction of ceramide. The novelty of this study is related to the use of the ASM-inhibitor imipramine to revert hypoglycemia-induced neuronal death.

The manuscript needs some improvements in order to be ready for publication, as described below.

It is already well known that agents that reduce ASM activity and thereby also ceramide levels tend to attenuate receptor-mediated apoptosis, stress stimuli-induced apoptosis, as well as growth factor-deprivation-mediated apoptosis and promote cell proliferation. Thus, ASM-inhibitors potentially have antiapoptotic and neuroprotective effects and have been extensively studied for the treatment of disorders such as brain ischemia, stroke, ethanol-induced neuronal cell death, Alzheimer’s dementia, Parkinson’s disease, Chorea Huntington, spinal cord injury, seizure disorder, glaucoma, and to protect against neurodegeneration occurring in multiple sclerosis. Specifically, it has been shown that therapeutic concentrations of the antidepressants amitriptyline and fluoxetine functionally inhibit Asm and reduce ceramide concentrations in the hippocampus (https://doi.org/10.1159/000442611). Curiously, none of these are mentioned in the introduction section or discussed, and it gives the wrong impression of a new idea. However, it has been well explored in the literature so far.

  1. Abstract needs to be improved. There are a lot of single sentences not connected. Avoid repeating “the present study” more than once.

<Response: We corrected it in the revised manuscript.>

  1. Review papers rather than original papers have been cited in most of the references in the Introduction and Discussion sections. Please use original articles.

<Response: We corrected it in the revised manuscript.>

  1. (Lane 25-26) – Please give some numerical data on diabetes/diabetic patients.

<Response: We added it in the revised manuscript.>

  1. (Line 27-29) - Please, explore the sentence, “During the last few decades, many diabetes patients have experienced hypoglycemia while attempting to tightly control their blood glucose to prevent diabetic complications.” How frequently does it happen? Or how many patients? Give some numerical data about the subject.

<Response: We appreciate this reviewer’s comment. According to Centers for Disease Control and Prevention (CDC), diabetic patients may experience low blood sugar as often as once or twice a week in the case of severe hypoglycemia (below 54mg/dL). Moreover, according to Alexandria Ratzki Leewing et at., as published in the 2018 BMJ journal, severe hypoglycemia was reported by 41.8% of all respondents, at an average rate of 2.5 events per person-year [ PMID:29713480].We added this content in the revised manuscript.>

  1. How to correlate repeated hypoglycemia (1st paragraph) with severe hypoglycemia (2nd paragraph)?

<Response: We appreciate this reviewer’s comment. This manuscript focuses only on severe hypoglycemia. Thus, as you mentioned, we deleted it.>

  1. (Line 36-38) – The statement “our lab has demonstrated that glucose reinstatement can cause an increase in neuronal death after hypoglycemia. Thus, we called it glucose reperfusion injury after hypoglycemia [10]” gives the wrong impression that authors research group described this classic pattern that was tested experimentally in a rat model of ischemia/reperfusion by Pulsinelli et al., several decades ago and repeated by several research groups. Please rephrase it.

<Response: We appreciate this reviewer’s comment. According to a paper published by Suh et al. in 2007, the glucose reperfusion performed after hypoglycemia further triggered hypoglycemic neuronal death. In addition, when conducting reperfusion, we only showed glucose, so we wrote “glucose reperfusion” to express it accurately. As the reviewer mentioned, we rephrased “glucose reperfusion injury” to “hypoglycemia/glucose reperfusion injury” in the revised manuscript.>

  1. (Line 43-46) - All sentences started in the same way. Please change it. The sentences need be integrated, telling a story. And not isolated sentences, as they are. It is mentioned three times that Sphingomyelin is found/ located/ component of cell membranes. Please, rewrite.

<Response: We corrected it in the revised manuscript.>

  1. (Line 50-57) - Same as above, all sentences started in the same way, and no connection between them. Please, rewrite.

<Response: We corrected it in the revised manuscript.>

  1. Please describe the “Animal Surgery and Severe Hypoglycemia Induction” keeping the order of the events: After overnight fasting, animals were anesthetized (description)…then a catheter was inserted… and so on….

<Response: We rewrote it in the revised manuscript.>

  1. (Line 93) – describe the anesthesia procedure: which anesthesia was used? How much was it used? For how long? Was there a maintenance dose?

<Response: We appreciate this reviewer’s comment. The rats were anesthetized with 2-3% isoflurane in 70% NO2/balanced O2 via nose cone until an isoelectric period (iso-EEG) appeared. When the iso-EEG appeared, we lowered anesthesia to 0.5-0.75% isoflurane, changed the ratio of the oxygen and nitrogen to 7:3, and then maintained it for 30 minutes. After 30 minutes of iso-EEG, during glucose reperfusion, we maintained 2% isoflurane for 2 hours. We added this contents in the Materials and Methods section>

11.When, after hypoglycemia induction, was IMP given intraperitoneally into the animals?

<Response: We appreciate this reviewer’s comment. As soon as Iso-EEG was finished, we injected imipramine. This content is in the procedure diagram of Figure 8.>

  1. What does the sentence below mean? “as time goes by, the sucrose solution permeates into the brain and touches the bottom, which means that the cryosection is ready for immunostaining.” Please, describe it correctly.

<Response: We rewrote it in the revised manuscript.>

  1. (Lines 247-248) – rephrase the sentence “FJB showed that we found large amounts of degenerating neurons...”

<Response: We corrected it in the revised manuscript.>

  1. What it is 4-HNE (4-hydroxy-2-nonenal)? Moreover, what does it do?

<Response: We appreciate this reviewer’s comment. 4-hydroxynoneal (4-HNE) is produced by lipid peroxidation in cells and is generally used as a marker of oxidative stress under pathological conditions. We added this content in the revised manuscript.>

  1. Grammar revision is necessary for section “3.7. Imipramine improves neurologic and cognitive function after hypoglycemia.”

<Response: We corrected it in the revised manuscript.>

  1. In Figure 7, do not repeat the text in the main manuscript. Please, correct.

The discussion section needs to be rewritten entirely. Most of the discussion section is a repetition of information already in the introduction section and repletion of results descriptions. Instead, results should be discussed based on their novelty and correlated with the literature. Figures should not be mentioned in this section either. There is no discussion at all.

<Response: We corrected it in the revised manuscript.>

  1. General English improvement.

<Response: We revised it and attached an English edit certificate. >

Round 2

Reviewer 2 Report

The authors did not answer the following points:

It is already well known that agents that reduce ASM activity and thereby also ceramide levels tend to attenuate receptor-mediated apoptosis, stress stimuli-induced apoptosis, as well as growth factor-deprivation-mediated apoptosis and promote cell proliferation. Thus, ASM-inhibitors potentially have antiapoptotic and neuroprotective effects and have been extensively studied for the treatment of disorders such as brain ischemia, stroke, ethanol-induced neuronal cell death, Alzheimer’s dementia, Parkinson’s disease, Chorea Huntington, spinal cord injury, seizure disorder, glaucoma, and to protect against neurodegeneration occurring in multiple sclerosis. Specifically, it has been shown that therapeutic concentrations of the antidepressants amitriptyline and fluoxetine functionally inhibit Asm and reduce ceramide concentrations in the hippocampus (https://doi.org/10.1159/000442611). Curiously, none of these are mentioned in the introduction section or discussed, and it gives the wrong impression of a new idea. However, it has been well explored in the literature so far. 

Review papers rather than original papers have been cited in most of the references in the Introduction and Discussion sections. Please use original articles.

Line 42-44 : the author lab did not demonstrate that "glucose reinstatement can cause an increase in neuronal death after hypoglycemia." This statement gives the wrong impression that authors research group described this classic pattern that was tested experimentally in a rat model of ischemia/reperfusion by Pulsinelli et al., several decades ago and repeated by several research groups. Also, the authors can not give a new name for the classic pattern as stated by the sentence: Thus, we called it “hypoglycemia/glucose reperfusion injury” after hypoglycemia [8]. The credits need to be given to the original authors.

The following sentence is wrong: "When the brain is initially in 30% sucrose, buoyancy causes the brain to float. Afterward, sucrose solution permeates into the brain tissue and the brain sinks to the bottom 2~3 days later, which means that the cryosection is ready for immunostaining." How is it possible that when the whole brain sinks to the bottom of sucrose solution cryosections would be ready? And the cryoprotection solution that the brain is immersed in before slicing? And how about the slicing itself? 

Line 251: cut off "and the" put a comma.

Line 341-350: As stated by the authors: "C18 and C24:1 exist in higher concentrations than other types, indicating that these two types play the most important roles in cells." Why is it important to identify the increase of C18 and C24:1 concentration caused by severe hypoglycemia? How can the decrease of C18 and C24:1 be related to neuroprotection? Please discuss, because the ceramide involvement in hippocampal neuronal death due to hypoglycemia has been reported before and is not new. Also, grammar correction is very necessary.

As I previously stated, most of the discussion section is a repetition of information already in the introduction section and a repletion of results descriptions. And there was not an improvement on it. 

As I also mentioned before: It is already well known that agents that reduce ASM activity and thereby also ceramide levels tend to attenuate receptor-mediated apoptosis, stress stimuli-induced apoptosis, as well as growth factor-deprivation-mediated apoptosis and promote cell proliferation. Thus, ASM-inhibitors potentially have antiapoptotic and neuroprotective effects and have been extensively studied for the treatment of disorders such as brain ischemia, stroke, ethanol-induced neuronal cell death, Alzheimer’s dementia, Parkinson’s disease, Chorea Huntington, spinal cord injury, seizure disorder, glaucoma, and to protect against neurodegeneration occurring in multiple sclerosis. Specifically, it has been shown that therapeutic concentrations of the antidepressants amitriptyline and fluoxetine functionally inhibit Asm and reduce ceramide concentrations in the hippocampus (https://doi.org/10.1159/000442611). There was no answer by the authors.

Author Response

It is already well known that agents that reduce ASM activity and thereby also ceramide levels tend to attenuate receptor-mediated apoptosis, stress stimuli-induced apoptosis, as well as growth factor-deprivation-mediated apoptosis and promote cell proliferation. Thus, ASM-inhibitors potentially have antiapoptotic and neuroprotective effects and have been extensively studied for the treatment of disorders such as brain ischemia, stroke, ethanol-induced neuronal cell death, Alzheimer’s dementia, Parkinson’s disease, Chorea Huntington, spinal cord injury, seizure disorder, glaucoma, and to protect against neurodegeneration occurring in multiple sclerosis. Specifically, it has been shown that therapeutic concentrations of the antidepressants amitriptyline and fluoxetine functionally inhibit Asm and reduce ceramide concentrations in the hippocampus (https://doi.org/10.1159/000442611). Curiously, none of these are mentioned in the introduction section or discussed, and it gives the wrong impression of a new idea. However, it has been well explored in the literature so far. 

  1. Review papers rather than original papers have been cited in most of the references in the Introduction and Discussion sections. Please use original articles.

<Response: we deleted references for review articles and replaced it with original articles.>

  1. Line 42-44: the author lab did not demonstrate that "glucose reinstatement can cause an increase in neuronal death after hypoglycemia." This statement gives the wrong impression that authors research group described this classic pattern that was tested experimentally in a rat model of ischemia/reperfusion by Pulsinelli et al., several decades ago and repeated by several research groups. Also, the authors cannot give a new name for the classic pattern as stated by the sentence: Thus, we called it “hypoglycemia/glucose reperfusion injury” after hypoglycemia [8]. The credits need to be given to the original authors.

< Response: We appreciate this reviewer’s comment. As you can see from the reference mentioned, Sang Won Suh, the corresponding author in this paper, first published in the journal of clinical investigation that glucose reperfusion causes more neuronal death after hypoglycemia {Suh, 2007, 17404617}. Therefore, we have continued to use this expression in our past papers, and we think the expression, “hypoglycemia/glucose reperfusion injury”, can be used based on the paper published by the corresponding author of this study.>

  1. The following sentence is wrong: "When the brain is initially in 30% sucrose, buoyancy causes the brain to float. Afterward, sucrose solution permeates into the brain tissue and the brain sinks to the bottom 2~3 days later, which means that the cryosection is ready for immunostaining." How is it possible that when the whole brain sinks to the bottom of sucrose solution cryosections would be ready? And the cryoprotection solution that the brain is immersed in before slicing? And how about the slicing itself? 

< Response: We appreciate this reviewer’s comment. When the brain sinks to toe bottom, it means that 30% sucrose has permeated the inside of the brain tissue. The reason why it is immersed in sucrose after being fixing with paraformaldehyde is the process of softening the tissue, and it is a pretreatment process that prevents the tissue from breaking during cryosection. Therefore, when sucrose seep into the brain tissue and brain sink to the bottom, the brain is placed on freezing media, frozen in the cryosection machine for 10minutes, and then the brain is sectioned. We briefly mentioned the above in the method section.>

  1. Line 251: cut off "and the" put a comma.

< Response: We corrected it in the revised manuscript.>

  1. Line 341-350: As stated by the authors: "C18 and C24:1 exist in higher concentrations than other types, indicating that these two types play the most important roles in cells." Why is it important to identify the increase of C18 and C24:1 concentration caused by severe hypoglycemia? How can the decrease of C18 and C24:1 be related to neuroprotection? Please discuss, because the ceramide involvement in hippocampal neuronal death due to hypoglycemia has been reported before and is not new. Also, grammar correction is very necessary.

< Response: We appreciate this reviewer’s comment.>

- First, we don’t understand clearly how can the decrease of C18 and C24:1 be related to neuroprotection. However, as mentioned in this study, we confirmed that C18 and C24:1 account for a large proportion of 11 ceramide types in the sham-operated group. After that, 11 types of ceramides tend to increase after severe hypoglycemia, of which C18 and C24:1 was significantly increased. In addition, we confirmed that C18 and C24:1 was significantly reduced when imipramine was treated. Therefore, in this study, we suggested that changes in concentration of C18 and C24:1 play an important role in hypoglycemia-induced neuronal death, and that C18 and C24:1 may be important targets for inhibiting neuronal death after hypoglycemia. But we also agree with the reviewer’s comments. So, as the reviewer mentioned, we think it is an interesting study to identify the detailed mechanisms which ceramide causes neuronal death in severe hypoglycemia by using a drug or compound that can specifically inhibit the production of C18 and C24:1. In the future, we will expand our research toward mechanisms. The above is mentioned in the discussion section.

- Second, we conducted this study after checking whether this study had been conducted before. There have been studies on the association between ceramide and neuronal death in other brain diseases such as ischemia, Parkin’s or Alzheimer’ disease, but no studies have been conducted on the association between ceramide and hippocampal neuronal death in severe hypoglycemia. Please let me know if there is any paper that has been studied like us before. So, we think this study is meaningful and new enough to show that inhibition of ceramide production through inhibition of ASMase can improve cognitive function aa well as suppress hippocampal neuronal death after injury.

- Third, we revised the paper and received English editing from an MDPI. We attached certification>

  1. As I previously stated, most of the discussion section is a repetition of information already in the introduction section and a repletion of results descriptions. And there was not an improvement on it. 

< Response: We appreciate this reviewer’s comment. So, we mentioned the contents of the introduction and results to a minimum, deleted a repetition of information and rewritten the discussion part as a whole.>

  1. As I also mentioned before: It is already well known that agents that reduce ASM activity and thereby also ceramide levels tend to attenuate receptor-mediated apoptosis, stress stimuli-induced apoptosis, as well as growth factor-deprivation-mediated apoptosis and promote cell proliferation. Thus, ASM-inhibitors potentially have antiapoptotic and neuroprotective effects and have been extensively studied for the treatment of disorders such as brain ischemia, stroke, ethanol-induced neuronal cell death, Alzheimer’s dementia, Parkinson’s disease, Chorea Huntington, spinal cord injury, seizure disorder, glaucoma, and to protect against neurodegeneration occurring in multiple sclerosis. Specifically, it has been shown that therapeutic concentrations of the antidepressants amitriptyline and fluoxetine functionally inhibit Asm and reduce ceramide concentrations in the hippocampus (https://doi.org/10.1159/000442611). There was no answer by the authors.

< Response: We appreciate this reviewer’s comment. >

- First, it has not been studied in severe hypoglycemia disorder. In addition, studies have shown that antidepressants have a neuroprotection effect, but these studies were not research es on hippocampal neuronal death and cognitive dysfunction. Our paper is different from previous studies in the disease model and animal used. In the paper mentioned (https://doi.org/10.1159/000442611), melatonin was injected twice a day into wild type and Asm-deficient mice (smpd1-/-) at a concentration of 10mg/kg for12 days. They showed ceramide concentration changes, oxidative stress, and neurogenesis, but did not confirm direct hippocampal neuronal death or cognitive dysfunction improvement. On the other hand, we used a rat 10 times larger than mice in our study and injected the same concentration of imipramine (10mg/kg, i.p) once a day for 1week. Of course, melatonin and imipramine are different drugs, but with short administration in larger animals at the same concentration, we directly showed the inhibitory effect of imipramine on hypoglycemia-induced neuronal death and also showed improvement in cognitive dysfunction.

- Second, we also checked the neuroprotection study through another antidepressant drugs mentioned by the reviewer. To select and explain some of the similar study to us, Yu Rong Guo et al. used mice and injected amitriptyline at a concentration of 10mg/kg (i.p), the same as imipramine, for 7 days and they showed just hippocampal neurogenesis {Guo, 2019, 31903069}. In the case of fluoxetine, previous studies used a concentration of 10~20mg/kg, and injected within a short period time like 30m, 3h, and 6h or for more than 9days after injury. These papers showed changes in neurogenesis or cortical neuronal death, and the disease model was either a chronic -stress model or an ischemia model, which was different from our study {Zhao, 2020, 32477152; Khodanovich, 2018, 29304004; Lim, 2009, 18855941}. Therefore, what is distinctly different from previous studies is that we revealed that imipramine has a neuroprotection effect at a smaller concentration than other antidepressant drugs, as well as providing the possibility of improving imipramine for cognitive dysfunction. In addition, we believe this study is meaningful enough because there has not been a study on the relationship between severe hypoglycemia-induced neuronal death and ASMase/ceramide pathway.